# COVID-19 Infection in Academic Dental Hospital Personnel; A Cross-Sectional Survey in Saudi Arabia

**DOI:** 10.3390/ijerph182010911

**Published:** 2021-10-17

**Authors:** Osama Abu-Hammad, Ahmad Alnazzawi, Hamzah Babkair, Safa Jambi, Maher Mirah, Ismail Abdouh, Rahaf Saeed Aljohani, Rahaf Ayeq, Layan Ghazi, Heba Al-subhi, Najla Dar-Odeh

**Affiliations:** 1College of Dentistry, Taibah University, Al Madinah Al Munawara 43353, Saudi Arabia; oabuhammad@taibahu.edu.sa (O.A.-H.); anazawi@taibahu.edu.sa (A.A.); Hbabkair@taibahu.edu.sa (H.B.); sjambi@taibahu.edu.sa (S.J.); mmirah@taibahu.edu.sa (M.M.); iabdouh@taibahu.edu.sa (I.A.); Rahaf.saeed2020@gmail.com (R.S.A.); Rahaf.ayman18@gmail.com (R.A.); layan.ghazi95@gmail.com (L.G.); alsubhi_hiba@hotmail.com (H.A.-s.); 2School of Dentistry, University of Jordan, Amman 11942, Jordan

**Keywords:** COVID-19, dental healthcare professionals, academia, pandemic, SARS CoV-2, long COVID, academic achievement

## Abstract

Background: Close patient contact is an essential component of clinical dental education, which can expose students and faculty to risk of COVID-19 and its sequelae. Methods: The study was a cross-sectional survey conducted among faculty and clinical students at an academic dental hospital in Al Madinah western Saudi Arabia. An online questionnaire was distributed to collect data on prevalence, risk factors, clinical manifestations, and long-term health and socioeconomic complications of COVID-19 infection. Results: Prevalence of COVID-19 was 19.6% among a total of 316 students and faculty. Participants cited family and friends as the primary source of infection (40.3%). Among cross-infection control practices, they cited failure to practice distancing as the primary reason for infection transmission (61.3%). The disease was symptomatic in 85.5% of infected personnel. Most frequently reported clinical manifestations were: fever, cough, malaise, and diarrhoea (74.1%, 56.5%, 40.3%, 32.3%, respectively). A proportion of 37.1% of infected personnel stated that they had long COVID-19, and 58.3% of infected students reported deteriorated academic achievement. Conclusions: One in five of clinical dental students and their faculty had COVID-19. Most cases were symptomatic, and a large proportion developed long COVID or adverse socioeconomic consequences. Regardless of the severity of symptoms encountered during the acute stage of COVID-19 infection, all infected dental healthcare personnel should be followed, especially those who report long COVID. Continuous follow-up and assistance for infected students may be warranted to mitigate the potential academic and mental drawbacks caused by the pandemic. Dental schools should adopt clear policies regarding COVID-19 transmission and prevention and should implement them in their infection-control education and training.

## 1. Introduction

It has been almost two years since the emergence of coronavirus disease-19 (COVID-19); however, unveiling the spectrum of its clinical characteristics and long-term complications seems to be a demanding task for clinicians and researchers across the globe. It was initially considered a viral respiratory infection manifested as pneumonia and associated with systemic manifestations similar to influenza. Initial reports identified fever, cough, fatigue, taste loss, dyspnoea, arthralgia, myalgia, and diarrhoea as the characteristic symptoms of the infection [1]. Pulmonary disease in the form of diffuse alveolar damage is the most common symptom detected in acute and late stages; however, several extrapulmonary manifestations are reported either due to the infection itself or due to the various therapeutic modalities used [2].

Perhaps the most agreeable aspect of COVID-19 among the scientific community worldwide is the highly contagious nature and rapid spread, which is challenging to members of all health professions. Dental healthcare professionals (DHCP) are potentially susceptible to this infection, having to work intimately with patients in the orofacial region. Furthermore, dental patients are at risk to contract infection either from DHCP themselves or other patients. Therefore, dental practitioners have to be aware of the various aspects of the disease, including its contagious nature and its impact on dental patients in terms of adverse oral health outcomes or safe delivery of oral healthcare services [2,3]. Many clinical findings have been reported for this infection; however, growing attention is currently being directed to the debilitating long-term complications known as long COVID or chronic COVID-19, which occurs in patients who develop one or more complications of the disease or continue to have the usual symptoms for longer than expected [4]. This period is generally considered to be four weeks after the acute illness [5].

Several articles on COVID-19 have addressed the disease impact on dental practice in general as well as its impact on mental and psychological well-being of dental students. However, there is scarce literature, if any, on COVID-19 prevalence, clinical characteristics, and long-term complications among DHCP, including students and faculty in academic dental institutions. Academic or teaching dental hospitals provide dental education that is essentially based on clinical practice and close patient contact. Safety and durability of the academic process should be adequately sustained to guarantee provision of successful clinical dental training. Dental clinical settings particularly teaching dental hospitals can be a serious source of infections if proper and solid cross-infection control policies are not implemented. Moreover, the role of dental students is not only confined to clinical duties, and provision of dental treatment to their patients. Their role exceeds that, taking on an educational nature by influencing attitudes of their patients towards promotion of their oral and general health behaviours [6].

Therefore, we conducted this study to determine the pattern of COVID-19 infection among dental students and faculty at an academic dental hospital located in western Saudi Arabia in terms of prevalence, clinical manifestations, and possible sources of infection. We also investigated the prevalence and types of long-term health and socioeconomic complications among infected personnel.

## 2. Materials and Methods

The study was a cross-sectional observational study conducted among dental faculty and students at Taibah University Dental Hospital located in Al Madinah, western Saudi Arabia. Collection of data was during March–August 2021. Inclusion criteria were clinical dental students (4th, 5th, 6th years, and interns) and faculty currently practicing at the hospital. Pre-clinical students and staff who were abroad during the period of the study were not invited and were excluded from the study. Sample size determination was carried out using the epidemiological software: Epi Info™ (CDC, Centres for Disease Control, Atlanta, GA, USA) based on a population size of 316 (number of students and staff who consented to participate). The expected frequency (outcome probability) was assumed to be 7.5% based on serological prevalence of SARS-CoV-2 among healthcare workers in Saudi Arabia [7]. A sample size of 61 participants was determined to provide 90% confidence level at 5% margin of error. The list of all students and faculty names and their contact details were obtained from the college administrative office. All eligible faculty and students at the hospital were invited to participate in this study.

Data were collected using an anonymous, online questionnaire created using Google Forms. A panel of three authors with expertise in cross-sectional surveys and epidemiological aspects of COVID-19 (O.A.-H., A.A., N.D.-O.) designed the first version of the questionnaire. It was composed of 19 closed-ended questions divided into three sections of demographics, COVID-19 clinical attributes, and outcomes. A pilot test was performed to ensure clarity of questions and reproducibility of responses. A group of five students and five faculty were invited to complete the questionnaire on two occasions separated by one week to compare responses. Face validity was carried out within the authors group who did not share in designing the questionnaire. Unclear or vague questions were modified. The calculated Cronbach alpha and Kappa values were considered acceptable (0.72 and 0.77, respectively). Subjects were contacted during working hours and invited to participate. Consenting participants completed the questionnaire in presence of co-investigators without interfering or influencing their responses.

Ethical approval was obtained from Taibah University College of Dentistry Ethics Committee, reference # TUCDREC/17012021/NDar-Odeh.

### Statistical Analysis

The Statistical Package for Social Science (SPSS) version 21 was used to calculate descriptives in the form of frequency, percentages, and prevalence. Statistical significance of association of demographic data with clinical attributes of the disease were calculated using cross-tabulation with chi-square test, with level of significance set at *p* ≤ 0.05.

## 3. Results

A total of 316 out of 344 eligible faculty and students participated in this study with a response rate of 91.9%. They were 143 (45.3%) males and 173 (54.7%) females. Mean age was 28.88 ± 8.77 years (range = 20–55 years). Participants less than 40 years old were 273 (86.4%), while those 40 years old or more were 43 (13.6%). Faculty were 98 (31.0%), and students were 218 (69.0%).

### Characteristics of COVID-19 among Infected Personnel

A total of 62 subjects (19.6%) stated that they had COVID-19 infection: 52 cases were diagnosed in 2020 and 10 cases in first half of 2021. Infections took place in the period starting from 30 March till 13 May 2021. Demographic characteristics and their cross-tabulation with infection status among participants are presented in Table 1. There were no statistically significant differences between infection status and variables of gender, age, and role in the hospital (*p* ≥ 0.5).

Among the 62 infected participants, one in three did not know the source of infection, and one in four thought that the dental hospital was the source. A substantial proportion thought that family and friends were the source (Table 2). Most infected participants thought that failure to practice distancing was the major factor in contracting the infection. Other factors cited were failure to properly use face masks or practice hand washing (Table 2). Only 7 (11.3%) of the infected sample did not practice quarantine, and five of those (71.4%) believed that they transmitted the infection to others (Table 2).

A total of 53 (85.5%) infected personnel reported symptomatic infection with fever, cough, malaise, and diarrhoea being the most commonly cited symptoms (Figure 1).

Average duration of signs and symptoms was 5.2 ± 4.2 days and ranged from 2–25 days, as described in Figure 2.

Long-term adverse health complications were reported by 24 (38.7%) infected personnel, and these were mostly respiratory and cardiovascular complications (Figure 3).

The reported socioeconomic sequelae were: psychological (*n* = 40, 64.5%) and economic (*n* = 9, 14.5%). Reduced academic achievement was also cited (*n* = 28, 58.3%) by infected students.

Participants < 40 years, students, and males were significantly associated with shorter duration of acute illness (*p* < 0.05), as shown in Table 3. On the other hand, males and students were significantly associated with reporting long-term complications (*p* < 0.05), as shown in Table 3.

## 4. Discussion

This study investigated prevalence and pattern of COVID-19 infection among DHCP in a geographic area known to be heavily impacted by the pandemic during its early phase in Saudi Arabia. During summer 2020, Saudi Arabia, one of the largest Eastern Mediterranean countries, was among the top 16 affected countries worldwide in terms of total numbers of COVID-19 cases, with a total of 6322 cases per million (6.3%) [8]. At that time, Al-Madinah, where this study was conducted, was among the top five Saudi regions with regards to incidence of COVID-19. Further, the dental hospital where the study was conducted is considered the only academic dental institution in the city, with an estimated 3000 patients attending per month [9]. It was important to assess whether the pandemic has a potential long-term influence on dental practice in general and dental education in particular. Most literature published so far has been concerned with perceptions and knowledge of DHCP on COVID-19 and methods advocated for mitigating risk of infection transmission among dentists and their patients [1].

A substantial proportion of COVID-19 infections approaching 20% among the study sample was noticed. This represents all cases of COVID-19 among the study sample that took place during 17 months from March 2020 till May 2021. There are scarce data on infection rate among DHCP. A previous study conducted in the USA reported a prevalence of only 2.6% positive COVID-19 tests among dentists [10]. A recent review reported an 11% prevalence of SARS-CoV-2 infection among frontline healthcare workers [11]. Furthermore, the prevalence in this study may be under-estimated because this disease could be subclinical in approximately 17.9–33.3% of infected subjects [12,13]. This is close to the percentage of asymptomatic cases found in this study (14.5%). On the other hand, Rivett et al. (2021) reported a higher percentage (57%) of asymptomatic cases among healthcare workers [14]. When comparing numbers of diagnosed cases in the years 2020 (April to December 2020) and 2021 (January to May 2021), it was noticed that there were 52 cases in 2020 and only 10 cases in the first half of the year 2021. This may reflect the reduced number of cases detected in the year 2021, probably as a result of initiating vaccination campaigns. In July 2020, Saudi Arabia ranked 14 among affected countries in the total number of cases [8]. One year later, there was much improvement, as the rank regressed to be 49 in the total number of cases worldwide [15].

Demographics had no association with rate of infection among the study sample who were all younger than 60 years. A higher rate of infection among females younger than 60 years was reported in the European union countries, and it was attributed to their larger representation in healthcare facilities [16]. The study sample here does not belong to frontline healthcare personnel. Another explanation can be based on the most frequently cited factor by participants as the source of infection, which was family and friends. Only one in four infected personnel considered the dental hospital as the source of infection. They also considered that the most important factor in transmitting the infection was failure to implement distancing, while inappropriate use of face mask was the least-cited factor. Dental practitioners are generally considered well prepared in cross-infection control measures, including the use of personal protective equipment (PPE), which has contributed to reducing infection rates [17]. Community-based studies, on the other hand, estimated that facemask use was more effective in reducing the risk of COVID-19 infection than distancing [18]. In the context of dental clinical settings, distancing is not applicable, as dental practice is largely carried out in close proximity to the patient, with high possibility of exposure to aerosols [19]. This highlights the importance of wearing the facemask and other PPE in controlling infection transmission [1].

The study showed that a minority of those infected still disregard recommended protocol in practicing quarantine. Infected individuals mostly believed that their failure to properly quarantine caused the transmission of infection to others. Quarantine is a crucial control measure in reducing community transmissions. However, the appropriate duration of quarantine is debatable due to the variability of incubation period and occurrence of asymptomatic cases [20]. Symptomatic cases in this study mostly manifested as fever, cough, malaise, and diarrhoea. This is consistent with other studies that showed that respiratory and gastrointestinal symptoms are the most common symptoms of COVID-19 infections [21]. It was noticed that males, young participants, and students had a significantly shorter duration of acute illness. This could be attributed to the stronger immunity anticipated among young and male patients [22].

All cases progressed to recovery with no fatalities; however, a substantial proportion of patients progressed to long-term complications or what is known as long COVID-19. The estimated proportion of patients who develop long COVID-19 ranges from 10–20% [23]. These complications seem to be higher in prevalence among our sample, and those were mostly respiratory and cardiovascular. Respiratory manifestations, such as chronic cough, and cardiovascular manifestations, such as acute coronary syndrome, are examples of the respiratory and cardiovascular complications reported in previous studies of COVID-19 patients [24]. It was noticed that long COVID-19 was significantly associated with male gender and student groups and not with the older age group. This is contrary to studies that reported an association between long COVID-19 and older age (≥40 years) [18] or female gender [22,25]. Older age was correlated with weaker immunity and hence higher susceptibility to long-term complications, while female gender was correlated with biological differences and the apparently higher tendency of females to reporting. In this study, the size of the older age group who had COVID-19 was relatively small. Further, cultural attitudes in reporting and genetic differences among ethnicities could contribute to the differences in influence of the gender factor found between this study and other studies conducted in different geographic areas. Regardless of age, gender, or any other factor, patients suffering long COVID-19 should be closely monitored, and proper healthcare services should be provided for as long as possible. When such patients are healthcare personnel, special attention should be paid in order to avoid crises in maintaining a sustainable and efficient workforce in health care services. Special programs should be designed for rehabilitation and management of “wounded healers” [23], especially when their role goes beyond clinical duties to include teaching and research in the academic settings. Of similar importance are the socioeconomic consequences of the pandemic. It was reported that first-line healthcare workers are at higher risk of experiencing adverse psychological outcomes in form of burnout, anxiety, fear of transmitting infection, feeling of incompatibility, and depression, among others [26]. Although DHCP do not usually work as first-line healthcare workers, literature suggests that COVID-19 patients may generally have a high burden of mental health problems, including depression, anxiety disorders, stress, panic attacks, and irrational anger, among other serious problems [27]. Most infected students stated that their academic achievement was adversely affected. Reviewing the literature revealed rich data on how the pandemic adversely influenced mental and academic performance of dental students. However, it seems that no studies have yet been conducted on students who are already infected with COVID-19. It is plausible to say that numerous factors may contribute to reduced academic achievement among infected students. These factors could be the long sick leaves, which may extend up to two weeks; the psychological effects of mandatory quarantine; and having to face the consequences of long COVID-19 for self and family among the extended families commonly found within Arab communities [8]. Nevertheless, it is expected that adverse factors in the academic environment can be properly identified and addressed, especially since university students in many parts of the world have shown sufficient knowledge and acceptance of COVID-19 vaccines [28].

It should be noted that no new cases of COVID-19 were reported among faculty and dental students in the study setting from the beginning of the academic year on 29 August 2021 so far (personal communication). This can potentially be attributed to the successful implementation of COVID-19 vaccination campaigns mandated by the government.

Our study has limitations. It was a single-centre study; however, the participating dental hospital was a teaching hospital dedicated to providing free dental services to a high flow of dental patients. This, together with a high response rate, is expected to provide relatively accurate data. Another limitation is represented by the unintended exclusion of participants older than 55 years since the age range of participants covered only young and middle-aged adults. This may result in a non-representative sample. It would be more informative in future studies to include older participants and those with chronic co-morbidities to investigate a possible effect of age and medical status on COVID-19 outcomes. On the other hand, the study participants were students and staff; therefore, it allowed to compare disease outcomes between these groups in addition to other variables of gender and age. To our knowledge, this is the first study to investigate clinical characteristics and complications of COVID-19 infection among a unique sample of dental students and their faculty. Future studies should aim to investigate long-term complications of COVID-19 in the academic clinical dental settings.

## 5. Conclusions

DHCP in an academic clinical setting had a substantial proportion of COVID-19 infections, which were subclinical in a minority of cases. One in four cases of infection were contracted at the dental hospital, probably due to failure to practice distancing. It is necessary to investigate reasons behind failure of some of the infected personnel to practice quarantine, particularly that this practice could be associated with infection transmission. Other aspects that need attention is the substantial number of patients who develop long COVID and the drawbacks that this pandemic has initiated among students, such as reduced academic achievement. Dental schools should adopt and announce clear policies regarding COVID-19 transmission and prevention and should implement them in their infection-control education and training.

## Figures and Tables

**Figure 1 ijerph-18-10911-f001:**
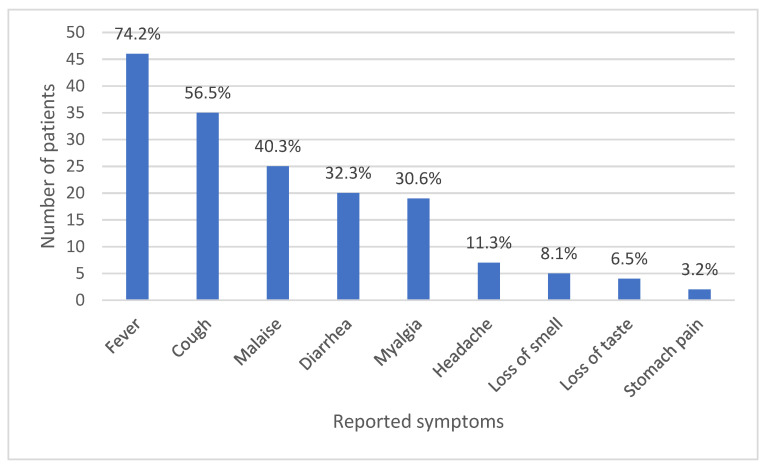
Acute symptoms of COVID-19 as reported by subjects contracting the infection.

**Figure 2 ijerph-18-10911-f002:**
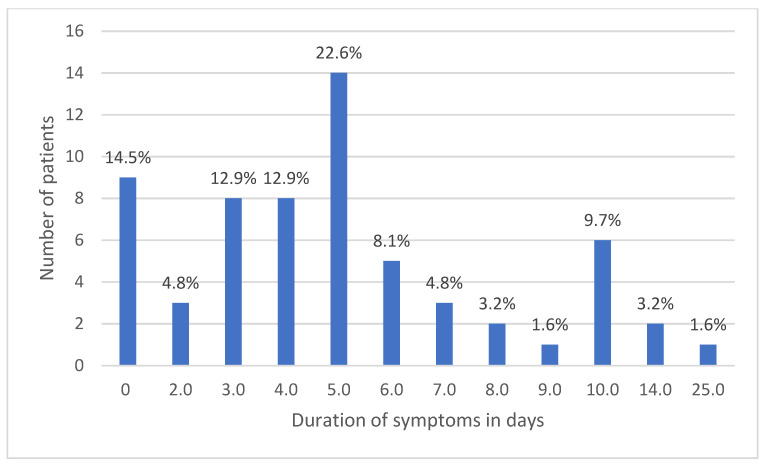
Duration of acute symptoms of COVID-19 (days) among infected personnel.

**Figure 3 ijerph-18-10911-f003:**
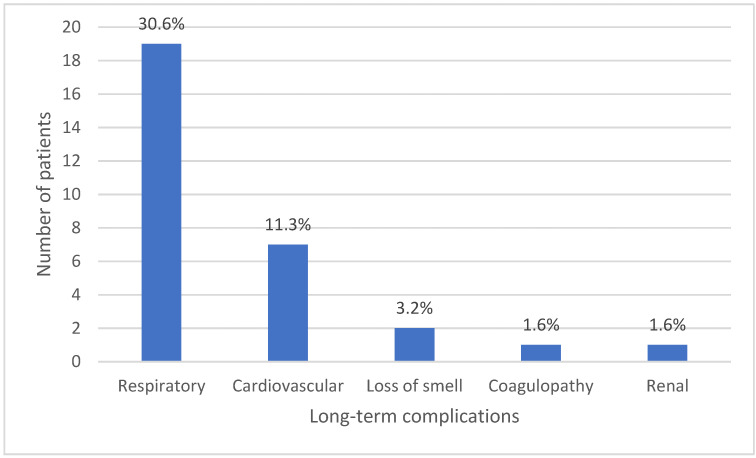
Types and frequency of long-term complications of COVID-19 among infected participants.

**Table 1 ijerph-18-10911-t001:** Socio-professional characteristics of participants cross-tabulated with COVID-19 infection status.

Socio-Professional Demographics	COVID-19 InfectionN (%)	*p*-Value
Gender		0.262
Females	30 (17.3%)	
Males	32 (22.4%)	
Age (years)		0.816
<40	53 (19.4%)	
≥40	9 (20.9%)	
Role in College		0.109
Faculty member	14 (14.3%)	
Student	48 (22.0%)	

**Table 2 ijerph-18-10911-t002:** Perceived sources of COVID-19 infection and improperly practiced cross-infection control practices as factors in infection transmission, as cited by the 62 individuals infected with COVID-19.

Infection Aspects	Frequency (%)
Source of infection	
Do not know	21 (33.9%)
Family/friends	25 (40.3%)
Dental hospital	15 (24.2%)
Marketplace	1 (1.6%)
Failure of proper cross infection control practices	
Face masks	10 (16.1%)
Hand washing	14 (22.6%)
Distancing	38 (61.3%)
Quarantine practiced	
Yes	55 (88.7%)
No	7 (11.3%)

**Table 3 ijerph-18-10911-t003:** Symptom duration in days and long-term complications/sequelae of COVID-19 infection cross-tabulated with age, gender, and role in college.

	Duration of Symptoms		Long-Term Complications	
	≤5 Days	>5 Days	No	Yes
**Age (years)**				
**<40**	39 (73.6%)	14 (26.4%)	31 (58.5%)	22 (41.5%)
**≥40**	3 (33.3%)	6 (66.7%)	7 (77.8%)	2 (22.2%)
***p*-value**	0.017		0.272	
**Gender**				
**Female**	15 (50.0%)	15 (50.0%)	25 (83.3%)	5 (16.7%)
**Male**	27 (84.4%)	5 (15.6%)	13 (40.6%)	19 (59.4%)
***p*-value**	0.004		0.001	
**Role groups**				
**Faculty member**	6 (42.9%)	8 (57.1%)	12 (85.7%)	2 (14.3%)
**Student**	36 (75.0%)	12 (25.0%)	26 (54.2%)	22 (45.8%)
***p*-value**	0.024		0.033	

## Data Availability

Data will be made available if requested from first author.

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
