# Peer review of "COVID-19 Infection in Academic Dental Hospital Personnel; A Cross-Sectional Survey in Saudi Arabia"

_ijerph, 2021, doi:10.3390/ijerph182010911_

Round 1

Reviewer 1 Report

This paper shows an analysis of data gathered through an online survey applied to students from an academic dental hospital.

Presenting results, discussion and conclusions based on the opinions from the survey and some assumptions.

in the whole document many statements are taken from citations to papers from the same authors, which i think that should be changed when possible.

Check the attached document for further comments please

Author Response

Dear reviewer,

Thank you for your valuable comments. All your comments and suggestions are addressed whether those mentioned below or those mentioned in the pdf file of the reviewed manuscript. The modifications are highlighted in blue. Please find below our responses in the red font.

Comments and Suggestions for Authors

This paper shows an analysis of data gathered through an online survey applied to students from an academic dental hospital.

Presenting results, discussion and conclusions based on the opinions from the survey and some assumptions.

in the whole document many statements are taken from citations to papers from the same authors, which i think that should be changed when possible.

Response: Statements referred to by the honorable reviewer were modified as requested.

Check the attached document for further comments please

Response: Responses to the comments in the pdf file are explained below

Abstract

What is the reason for this conclusion? among all academic dental personnel?

Response:  This was modified to A substantial proportion of dental students and their faculty had COVID-19. Also, the conclusions section was modified.

Are there any statistics to support this?

Response: This statement describes the potential susceptibility of DHCP to COVID-19 and other respiratory infections. We are not attempting to say that there is a high prevalence of this infection among the group of DHCP. Actually, reports on the prevalence rate are scarce. Therefore, we changed the word “particularly susceptible” to “potentially susceptible”. We hope that this removed any ambiguity related to this statement.

citation appears to be out of place

Response: Citation was removed

Results

This appears to be a description of the dataset and should be in the materials and methods section

Response: We completely understand your point of view as there are two schools regarding reporting the total sample participating in the study and their social characteristics. One school prefers to include this in methods section, and the other school, that we use so far in our publications, include this in the results section. We included this in the results section because we think that methods section should include only methods that can be used repeatedly with the same validity. While these results may be variable between studies. However, we are willing to include them in the methods section if you find this a necessity.

Clarify the metric (i.e. days):

Response: Metric (days) is now included in the figure, and its caption

Borader explanation of this FIgure is required, people should be able to understand its content from the caption... it is not clear what does the blue bar mean, what about the orange bar? why is it negative? Caption should explain briefly all of this. This applies to all Figures and tables

Response:

The figure now is more explanatory, and it is modified so that there is only blue color. Captions of all tables and figures are now modified to be more explanatory.

Discussion

Second paragraph

Is there science supporting this claim? citation required

Response: This statement was used based on the article written by Michael Day in The BMJ early 2020 when he referred to the work of an Italian academic who “claimed striking evidence that most people infected with covid-19 show no symptoms but are still able to infect others” (Day M. Covid-19: identifying and isolating asymptomatic people helped eliminate virus in Italian village BMJ 2020; 368 :m1165 doi:10.1136/bmj.m1165)

However, more recent research now reports lesser proportions of asymptomatic patients. For example two studies report a range of 17.9%-33.3%. ( [12] Mizumoto K, Kagaya K, Zarebski A, Chowell G. Estimating the asymptomatic proportion of coronavirus disease 2019 (COVID-19) cases on board the Diamond Princess cruise ship, Yokohama, Japan, 2020. Euro Surveill. 2020 Mar;25(10), [13] Nishiura H, Kobayashi T, Miyama T, Suzuki A, Jung SM, Hayashi K, Kinoshita R, Yang Y, Yuan B, Akhmetzhanov AR, Lin-ton NM. Estimation of the asymptomatic ratio of novel coronavirus infections (COVID-19). Int J Infect Dis. 2020 May;94:154-155.

While Rivett et al (2020) report a higher percentage of 57% among healthcare personnel. We modified the statement according to your comment and added citations (12-14) so as to conform to the most recent data.

Page 8

rephrase to avoid using apostrophes

Response: The sentence was rephrased

I see the need to dig deeper into the conclusions, make a larger dissertation about the results found and their implication. I mean, these conclusions could be stated without the research made, and are true almost in any medical field. Your conclusions should justify your research

this word should be included in the conclusions of the abstract when talking about the prevalence in the conclusion

Response: Both conclusion sections (abstract and text) were modified according to the above comment

Reviewer 2 Report

Thank you for the opportunity to review this paper. It is demonstrated that heath care workers (HCWs) were among the most involved categories in COVID 19 for both workload and level of infection contracted, and especially dental personnel. Moreover, previous studies have demonstrated how prevention and control measures are evolving, however adherence to them are essential for the final success. In this context, the paper under review is aimed at investigate prevalence, risk factors, clinical manifestations and long-term clinical and socioeconomic complications of COVID-19 among those who were infected.

The article is interesting and may provide important information for public health, but it must be improved. Moreover, it seems part of the same study whose results are reported in an other paper that I am already reviewing for Vaccines. Therefore, the suggestions relate to the general protocol of the study and paper are the same.

Title: it is overstated and the object of the study must be better explained, moreover the geographical area studied must be declared.

Introduction: The authors should make it clear about what is the gap in the literature that is filled with this study? What is the contribution of the study to the literature? What are the implications of the study?

Methods: The reliability of the questionnaire was evaluated, but what about face validity and intelligibility? Statistical analysis: I suggest to insert a measure of the magnitude of the effect for the comparisons. Please consider to include effect sizes.

Discussion: I also suggest expanding. Emphasize the contribution of the study to the literature, the implications and recommendations based on previous experience also enlarging the concept reported in the conclusion “More efforts are required to improve awareness on transmission and methods of prevention”: at the moment the vaccination campaign is the first method to counteract the COVID-19, therefore student’s Knowledge and Acceptance of COVID-19 Vaccination and related effectiveness of the information strategy must be discussed (refer to Gallè, F. et al Knowledge and Acceptance of COVID-19 Vaccination among Undergraduate Students from Central and Southern Italy. Vaccines 2021, 9, 638). Limits section must be improved.

English must be improved.

Author Response

Dear Reviewer,

Many thanks for your valuable comments. All your comments and suggestions were addressed and the text was modified accordingly. Modified areas are highlighted in yellow. Please find below our responses in the red font.

Thank you for the opportunity to review this paper. It is demonstrated that heath care workers (HCWs) were among the most involved categories in COVID 19 for both workload and level of infection contracted, and especially dental personnel. Moreover, previous studies have demonstrated how prevention and control measures are evolving, however adherence to them are essential for the final success. In this context, the paper under review is aimed at investigate prevalence, risk factors, clinical manifestations and long-term clinical and socioeconomic complications of COVID-19 among those who were infected.

The article is interesting and may provide important information for public health, but it must be improved. Moreover, it seems part of the same study whose results are reported in an other paper that I am already reviewing for Vaccines. Therefore, the suggestions relate to the general protocol of the study and paper are the same.

 Response: Many thanks for your valuable comments. We addressed all your comments, and all modified areas of the manuscript are highlighted in yellow.

Title: it is overstated and the object of the study must be better explained, moreover the geographical area studied must be declared.

 Response: The title is now modified and the geographical area is stated in the title.

Introduction: The authors should make it clear about what is the gap in the literature that is filled with this study? What is the contribution of the study to the literature? What are the implications of the study?

 Response: Introduction is now modified to address the above points.

Methods: The reliability of the questionnaire was evaluated, but what about face validity and intelligibility?

 Response: Regarding face validity.  Actually, face validity was carried out by the authors.  Authors who were assigned to carry out this role are experts with sufficient previous experience.  We added some sentences to the methods section to clarify this.

Statistical analysis: I suggest to insert a measure of the magnitude of the effect for the comparisons. Please consider to include effect sizes.

Response: Effect size or Cohen’s d requires the calculation of means and standard deviations of 2 groups.  However, we cannot calculate means and standard deviations in our study because our data are not continuous scale data, our data in fact are dichotomous nominal data

Discussion: I also suggest expanding. Emphasize the contribution of the study to the literature, the implications and recommendations based on previous experience also enlarging the concept reported in the conclusion “More efforts are required to improve awareness on transmission and methods of prevention”: at the moment the vaccination campaign is the first method to counteract the COVID-19, therefore student’s Knowledge and Acceptance of COVID-19 Vaccination and related effectiveness of the information strategy must be discussed (refer to Gallè, F. et al Knowledge and Acceptance of COVID-19 Vaccination among Undergraduate Students from Central and Southern Italy. Vaccines 2021, 9, 638).

Response:

Discussion section was modified to address the above points. We cited the above valuable reference to explain in the influence of knowledge and positive vaccine attitudes on curbing infection rates.

Limits section must be improved.

Response: Study limitations are explained further in the discussion

English must be improved.

Response: The article was revised for English.

Reviewer 3 Report

Figure 1. Symptoms as reported by subject contracting the infection.

Figure 2. Duration of symptoms among infected personnel.

Figure 3. Long-term complications among infected participants.

  1. Please add % percentage on the Fig.1~Fig.3.
  2. There is orange color on Fig.3, please check.
  3. Where is Table 3?
  4. What / how is the definition of shorter duration (5 days) of acute illness in Table 4?

Author Response

Dear reviewer,

Many thanks for your valuable comments.

Comments and Suggestions for Authors

Figure 1. Symptoms as reported by subject contracting the infection.

Figure 2. Duration of symptoms among infected personnel.

Figure 3. Long-term complications among infected participants.

  1. Please add % percentage on the Fig.1~Fig.3.

Response: Percentages now appear in figures 1-3

  1. There is orange color on Fig.3, please check.

Response: Figure 3 was revised and only blue color appears

  1. Where is Table 3?

Response: We apologize for this mistake.  There are three tables only, and table 4 in the original version should be referred to as table 3. Now this is corrected, and highlighted in green.

  1. What / how is the definition of shorter duration (5 days) of acute illness in Table 4?

Response: Average duration of signs and symptoms in this study was 5.2±4.2 days and it ranged from 2-25 days. Therefore, a duration of 5 days was considered the cut point for comparison in table 3. However, based on the duration of symptoms (all were less than one month) these symptoms were considered acute (figure 2).

Reviewer 4 Report

My first question is on the questionnaire tool used? Was the tool adapted from a specific study ? If yes please aknowledge the study / studies?

Secondly if it was adapted from another study, provide a clear justification of why you needed to measure the Chrobach's  levels?

Thirdly you state that your prevalence of 19% is high, on what basis? what is your benchmark? Can you also benchmark with data from a country that has similar socio-economic level. I did not think it was appropriate to benchmark with the USA.

Another question, could it be that had you collected the data in Winter, your prevalence would have been much higher? Something to deliberate on in discussion.

Lastly can you provide us with a few sentences on the limitations of your study.

Author Response

Dear Reviewer,

Many thanks for your valuable comments. All your comments were addressed in the revised version of the manuscript, and highlighted in pink. Please find below our responses in red font.

My first question is on the questionnaire tool used? Was the tool adapted from a specific study ? If yes please aknowledge the study / studies?

Response: A panel of three authors with expertise in cross-sectional surveys and epidemiological aspects of COVID-19 (O.A-H., A. A., N. D-O) designed the first version of the questionnaire. A pilot test was performed to ensure clarity of questions and reproducibility of responses. A group of five students and five faculty were invited to complete the questionnaire on two occasions separated by one week to compare responses.  Face validity was carried out within the authors group who did not share in designing the questions for the questionnaire.  Unclear or vague questions were modified. The calculated Cronbach alpha and Kappa values were considered acceptable (0.72 and 0.77 respectively).

The above paragraph explains the process of designing and testing the questionnaire as described in the methods section of the manuscript. Based on your query we added the sentence on designing the questionnaire to confirm that it was not adapted from other studies.

Secondly if it was adapted from another study, provide a clear justification of why you needed to measure the Chrobach's  levels?

Response: As explained in the previous response, the questionnaire was designed by our group, that is why Chronbach alpha was measured.

Thirdly you state that your prevalence of 19% is high, on what basis? what is your benchmark?

Response: All expressions referring to the prevalence of COVID-19 within the study sample were modified to “substantial proportion”, to avoid any ambiguity related to the word “high prevalence”.

In Saudi Arabia the prevalence of COVID-19 before the introduction of vaccines was around 6%. Also, the overall national serological prevalence of COVID-19 in Saudi Arabia was 11% according to a recent study published in July this year. (ref #7) Although a prevalence of 19.6% seems to be relatively high, we removed the expression “high prevalence” and replaced it with “substantial proportion” throughout the text.

Can you also benchmark with data from a country that has similar socio-economic level. I did not think it was appropriate to benchmark with the USA.

Response: We totally agree with the honorable reviewer that one should be careful in choosing the benchmarks. Up to our best knowledge, so far, no studies were conducted among similar samples of healthcare workers, at least in countries with similar ethnic and geographic background. As for the socioeconomic level, both Saudi Arabia and United States are categorized by the World Bank as high income countries (https://data.worldbank.org/country/XD, accessed 6 October 6, 2021). Although we acknowledge the possibility of differences in implementation of healthcare policies between the two countries, we have to confirm that Saudi Arabia has completely implemented strategies and policies related to COVID-19 pandemic in healthcare and higher education in concordance with those designed by international authorities including the WHO, and CDC. This applies also to the vaccines policies.  

Another question, could it be that had you collected the data in Winter, your prevalence would have been much higher? Something to deliberate on in discussion.

Response: The questionnaire was distributed during March-August 2021, however, it collected data on “ever” infection with COVID-19 (that included staff who were previously infected during last year). According to the results section “Infections took place in the period starting from 30/3/2020 till 13/5/2021.” Therefore, all cases of infections affecting participants were documented whether those occurring in winter or other seasons. We should explain, though, that according to the mandatory vaccine policy adopted by the government, all our staff and students are now vaccinated. So far, no new cases of COVID-19 were reported among staff and students from the beginning of the academic year which started on August 29 till the time of writing this revision (8 October 2021). This is now explained in the discussion section (highlighted in pink)

Lastly can you provide us with a few sentences on the limitations of your study.

Response: A limitations section now appears in discussion (highlighted in yellow)

Round 2

Reviewer 1 Report

I have nothing to add or comment.

Thank you

Reviewer 2 Report

The paper was improved according to the suggestions received, in my opinion it is suitable for publication